# Personalized Metabolic Avatar: A Data Driven Model of Metabolism for Weight Variation Forecasting and Diet Plan Evaluation

**DOI:** 10.3390/nu14173520

**Published:** 2022-08-26

**Authors:** Alessio Abeltino, Giada Bianchetti, Cassandra Serantoni, Cosimo Federico Ardito, Daniele Malta, Marco De Spirito, Giuseppe Maulucci

**Affiliations:** 1Neuroscience Department, Biophysics Section, Università Cattolica del Sacro Cuore, 00168 Rome, Italy; 2Fondazione Policlinico Universitario A. Gemelli IRCSS, 00168 Rome, Italy; 3RAN Innovation, Viale della Piramide Cestia, 00153 Rome, Italy

**Keywords:** metabolism, deep learning, gated recurrent unit, wearables, forecasting, diet plans, digital nutrition

## Abstract

Development of predictive computational models of metabolism through mechanistic models is complex and resource demanding, and their personalization remains challenging. Data-driven models of human metabolism would constitute a reliable, fast, and continuously updating model for predictive analytics. Wearable devices, such as smart bands and impedance balances, allow the real time and remote monitoring of physiological parameters, providing for a flux of data carrying information on user metabolism. Here, we developed a data-driven model of end-user metabolism, the Personalized Metabolic Avatar (PMA), to estimate its personalized reactions to diets. PMA consists of a gated recurrent unit (GRU) deep learning model trained to forecast personalized weight variations according to macronutrient composition and daily energy balance. The model can perform simulations and evaluation of diet plans, allowing the definition of tailored goals for achieving ideal weight. This approach can provide the correct clues to empower citizens with scientific knowledge, augmenting their self-awareness with the aim to achieve long-lasting results in pursuing a healthy lifestyle.

## 1. Introduction

The global obesity epidemic has been spreading throughout most countries since the 1980s. Obesity contributes directly to incident cardiovascular risk factors, including dyslipidemia, type 2 diabetes, hypertension, and sleep disorders [1,2,3]. Obesity also leads to the development of cardiovascular diseases independently of other cardiovascular risk factors. More recent data highlight abdominal obesity, as determined by waist circumference, as a cardiovascular disease risk marker that is independent of body mass index [4,5]. Lifestyle modification and subsequent weight loss improve both metabolic syndrome and associated systemic inflammation and endothelial dysfunction, leading to a reduction of coronary artery disease, heart failure, and atrial fibrillation [6,7,8,9].

Quantifying lifestyle modifications to decrease cardiovascular risk is nowadays conceivable following the increased use of wearable devices, such as smartwatches, smart bands and impedance balances. These devices allow the real-time and remote monitoring of physiological parameters. As measurement and feedback systems become more refined and personalized, these devices can help people to change their lifestyles and improve wellbeing. Moreover, they have the potential to be linked into a wide range of lifestyle support services through community, public and private providers. An important improvement in managing the huge variety of wearable and portable devices comes from web-based applications. Several solutions exist on digital stores, but they mostly suffer from incomplete and not well-defined food databases and lack of personalization due to the scarce integration of the information flux: users must rely on different applications, furnishing partial and unrelated information about their metabolic state, where energy intake and expenditure are not directly related. To overcome this issue, we developed a digital web-based application (ArMOnIA) integrating dietary, anthropometric, and physical activity data [10]. Data flows from smart devices (smart band and impedance balance) and diet diaries are collected to build an accurate and personalized estimation of energy balance (accounting for individual body composition, age, and hydration state). We already showed, in a single-arm uncontrolled prospective study on self-monitored voluntary normal or overweight adults, that this application, by simply allowing the visualization of the energy balance in a dashboard, helps users to significantly decrease their average energy balance and consequently BMI in a period of 45 days [10]. The data streams provided by this platform can be analyzed relying on machine learning and artificial neural networks, with the aim to provide predictive and personalizable computational models of metabolism. In particular, the problem of the prediction of weight variations traditionally relies on estimations based on thermodynamic models depending on age, height, gender, and current weight [11]. However, diet predictors developed through these models have limited application because they assume weight stability and do not account for factors such as microbiome, variations in type and expression of genes linked to nutrition, and quality and quantity of physical activity. Some human genome-scale metabolic models (GEM), such as Recon3D [12], contain the human gene–protein reaction associations and can mechanistically predict metabolic fluxes. However, these complex models need long elaboration times or high-performance computing (HPC) and cannot be embedded in edge computing (EC) to improve scalability and performance. Moreover, the personalization of metabolic models remains challenging [13], as they require new methodological approaches to integrate molecular and physiological data. Data-driven models of human metabolism would constitute a reliable, fast, and continuously updating model for predictive analytics. These models could indeed offer crucial data for achieving the best weight forecasts and the creation of individualized diet and exercise plans. Differently from the well-established knowledge-driven models, data-driven models can account for all of the metabolic processes, from genetic predispositions to current microbiome composition, affecting weight changes. Relying on this information embedded in the model, they could provide for personalized weight forecasts and for the creation of individualized diet and exercise plans, with the aim to achieve long-lasting results in pursuing a healthy lifestyle. To this aim, here we developed a personalized model of end-user metabolism, the Personalized Metabolic Avatar (PMA), to estimate its reactions to diets. PMA consists of a gated recurrent unit (GRU) deep neural network allowing the prediction and simulation of personalized weight variations according to macronutrient composition [14,15,16] and daily energy balance [17] and allowing the generation of tailored diet plans. PMA may be adopted to gradually improve adoption of healthy habits in a person-specific fashion.

## 2. Materials and Methods

### 2.1. Study Population and Protocol

In this single-arm uncontrolled prospective study, a group of four adult volunteers (three normal and one overweight) recruited from our lab staff self-monitored daily their weight, diet and step count for more than 300 days using the ArMOnIA app, without predetermined objectives or intervention. Other assessment data were collected in-person via digital diaries. The four participants shared their personal data after signing an informed consent. The protocol is as follows:
*Food diaries*: users must register daily the foods eaten during breakfast, lunch, dinner and snacks.*Physical activities (PA):* users must wear a smart band all day and all night, especially during physical activities where they have to specify the type of activity performed. These include: jogging, walking, swimming, working out, general sports, etc. Whenever participants forget to track their own activities with the smart band, they must register them into the ArMOnIA app, where the calories burned from these activities are evaluated through the compendium [18]. This is also performed for other activities not monitored by the smart band, such as house cleaning, driving, etc.*Weight monitoring:* users have to weigh themselves barefoot every day after waking up using an impedentiometric balance.


### 2.2. Wearables and Devices

The following devices were chosen for tracking anthropometric and PA data:
MiBand 6, a smart band (Xiaomi Inc.^®^, Beijing, China), for tracking PA and estimating calories burned during exercises (walking, running, etc.).Mi Body Composition Scale, an impedance balance (Xiaomi Inc.^®^, Beijing, China), for tracking anthropometric data such as: weight, resting metabolism, fat rate, muscle rate, bone mass.


These devices already had been used in three studies on PubMed, and 11 clinical trials had been performed using MiBand-1. Validation results in estimating RMR can be retrieved in a recent publication [19].

### 2.3. Data Collection, Storage and Retrieval through an Ad Hoc Developed Web App and Estimation of Personalized Energy Balance

A web application (www.apparmonia.com, accessed 21 July 2022) was developed in Python 3.8 with the libraries Django (https://www.djangoproject.com/, accessed 21 July 2022) and Django_plotly_dash (https://django-plotly-dash.readthedocs.io/en/latest/, accessed 21 July 2022) for data collection, storage, and visualization of energy balance through a dashboard [17].

The web application allows for data collection, storage, analysis and visualization. These are detailed below.

#### 2.3.1. Data Collection

Data provided in-person through a digital diary: food and other activities not included in the smart band (home activities, music playing, driving, etc.).

Data from the smart band and impedance balance were retrieved through the ZEPP Life^®^ app (Anhui Huami Information Technology Co., Ltd., Hefei, China).

#### 2.3.2. Data Storage

Retrieved data underwent anonymization and are then stored into a NoSQL database (MongoDB^®^, New York, NY, USA, https://www.mongodb.com/, accessed 30 June2022).

#### 2.3.3. Data Retrieval

The quantities retrieved from the database needed for the development of PMA were the following:
*w* is the weight acquired daily by the Mi Body Composition Scale.*m_C_* is the mass expressed in grams of total daily carbohydrate intake, *m_L_* is the mass expressed in grams of total daily lipid intake, and *m_P_* is the mass expressed in grams of total daily protein intake.daily energy balance, *EB*, calculated according to the formula

*EB = EI − TEE*,(1)
where *EI* is the daily energy intake, and *TEE* is the daily total energy expenditure.

*EI* is considered as the sum of all ingested calories as retrieved from the following databases: DIETABIT (www.dietabit.it, accessed 5 July 2022), CREA (www.crea.gov.it, accessed 5 July 2022), BDA (www.bda-ieo.it, accessed 5 July 2022), and OPENFOODFACTS (www.it.openfoodfacts.org, accessed 5 July 2022).

*TEE* is calculated according to the formula
*TEE = RMR + TEA + TEF*(2)
where *TEA* is the thermic effect of activity, *RMR* is the resting metabolism ratio, both measured using the values provided by the ZEPP Life^®^ app [19], and *TEF* is the thermic effect of food, referring to the energy expenditure related to food consumption [20] (i.e., digestion, absorption, assimilation, and storage), dependent on the amount and type of food consumed, which accounts for about 10% [21] of *TEE* and is estimated from food data through the following formula:(3)$$TEF=0.095\cdot \left(m_{c}\cdot 3.75\right)+0.015\cdot \left(m_{L}\cdot 9\right)+0.25\cdot \left(m_{P}\cdot 4\right)$$

### 2.4. Data Preprocessing

We considered energy balance and food composition as the main drivers of weight variations [10,22]. As already introduced in Section 2.3, the datasets used for the construction and testing of the model consisted of the following data:
Weight: *w(t)* [kg]Energy balance: *EB(t)* [kcal]Daily carbohydrate intake: $m_{c}$*(t)* [g]Daily protein intake: $m_{p}$*(t)* [g]Daily lipid intake: $m_{l}$*(t)* [g]Week cosine: cos($\frac{2}{7}\pi t$)Week sine: sin($\frac{2}{7}\pi t$)


In Figure 1A,B, sample *w(t)*, *EB(t)* and $m_{c}$*(t),*$\;m_{p}$*(t),*$\;m_{l}$*(t)* time series are reported. The last two terms, *week cosine* and *week sine*, were introduced to account for seasonality that can affect diet and PA habits, as shown in previous studies [23]. So far, in Appendix A, we showed a violin plot of a representative user reporting the distribution of the energy balance through all days in a week. As we can see, there is a variation among days confirmed also by statistical tests (Appendix A).

We then handled missing values (below 3% of the entire dataset) using the ‘*pad*’ method, taking values from the previous row. During imputing, test and train were separated to avoid crosstalk between the two sets.

*EB(t)* values can be affected by biases due to wrong insertion of food quantities, which are typically underestimated [24]. To account for these biases, we calculated for each time point the weekly variation of $EB_{week}\left(t\right)$ and the weekly weight variation of $\Delta w_{week}\left(t\right)$ and fitted with a linear regression model $EB_{week}=a\cdot \Delta w_{week}+b$ (Appendix A). *b* is the average bias in the estimation of the energy balance, which was subtracted from the estimated *EB(t)*.

### 2.5. PMA Development with RNN Network

PMA was shaped as output of a deep recurrent neural network, bridging the evolution of weight w(t) and $m_{c}$*(t),*$\;m_{p}$*(t),*$\;m_{l}$*(t)* (Section 2.4). Recurrent neural networks (RNN) are a very flexible class of neural networks, widely used to solve problems involving dependent data, such as time series. Therefore, this type of neural network best suited our application needs. Among the RNNs, we selected the mono-layer GRU (Appendix A [25]).

#### Data Preparation

Before deep learning can be used, time series forecasting problems must be re-framed as supervised learning problems. It is standard practice to use lagged observations (e.g., *t* − 1) as input variables to forecast the current time step (*t*). This is called ‘*multi-step forecasting*’ [26]. Calling *k* the lagged observation, the supervised learning dataset is reframed as:$$var_{1}\left(t-k\right)\;\ldots \;var_{7}\left(t-k\right)\;\ldots \;var_{1}\left(t-i\right)\;\ldots \;var_{6}\left(t-1\right)\;\ldots \;var_{7}\left(t-1\right)\;\ldots \;var_{1}\left(t\right)$$
where the overall time series are renamed with the string $var_{j}$, where *j* indicates the variable considered running from 1 to 7.

### 2.6. Model Selection

The first step in the development of the PMA is the definition of the architecture of the RNN used.

For this work, the most suitable architecture found for our application (Appendix A) was composed of the following layers:
*Input layer*: weight and exogenous series such as EB and food composition (carbohydrate, protein and lipid content expressed in grams) at previous times with respect to the output (plus historical values from the time series target). This corresponds to the
$x_{t}$ of Equation (S4), defined as follows: $x_{t}=[EB\left(t-k\right),$ $m_{c}\left(t-k\right)$, $m_{p}\left(t-k\right)$, $m_{c}\left(t-k\right)$, w(t−k), …, EB(t−1), $m_{c}\left(t-1\right)$, $m_{p}\left(t-1\right)$, $m_{c}\left(t-1\right)$, with *k* the lagged observation (specific for each user, as explained below).*Hidden layer*: a GRU neural network with the addition of a dropout layer (Appendix A [27,28]).*Output layer:* composed of one output, the weight *w* (*t* + 1) at time *t* + 1.


After that, we needed to choose the best set of hyperparameters (HP) to allow the model to predict accurately for every dataset used.

HP are the number of neurons, the type of activation function, the batch size, the number of epochs, the dropout value and the lookback value *k* (how many time steps we look back for the forecasting of the time series target). HP tuning was carried out to find the possible best sets to build the model from a specific dataset and with a specific goal [29]. HP tuning consists of the scanning of macro-parameters for the reduction of a loss function. Typically, in time series forecasting, the tuning is carried out to reduce the root mean squared error (RMSE) of the test-train forecasting (see Equation (4) below). Nevertheless, for our study, we introduced several constraints in selecting HP to guarantee correct dynamics of weight variations. In order to do so, we performed a simulation for 7 days (described in Section 2.7), considering diet plans consisting of different *EB* values: −1000,−500, 0, 500 and 1000 kcal. HP that did not respect the following conditions were discarded:
w(t+7)−w(t)>0 for EB=1000 kcalw(t+7)−w(t)<0 for EB=−1000 kcal
$w{\left(t+7\right)}_{EB=1000}-w{\left(t+7\right)}_{EB=-1000}<10\;\mathrm{kg}$w(t+7)−w(t) has to be an increasing function of *EB*


After this preselection, the choice of the best set of parameters was then made through minimization of the RMSE of the test-train forecasting, evaluated according to the formula:(4)$$\mathrm{RMSE}=\sqrt{\overset{n}{\underset{i=1}{\sum }}\frac{{\left(\hat{y_{i}}-y_{i}\right)}^{2}}{n}}$$

In the following, we report in detail the HP parameter scanning sets:

*Number of neurons*: The number of neurons in the hidden layer for the GRU neural network has to be adjusted to the solution complexity: the task with a more complex level to predict needs more neurons. To consider GRU with increasing complexity, the number of neurons was chosen from the following range: 50, 100, 150 and 200.

*Activation function*: The activation function of the GRU mono-layer is crucial to compute the input values into output values. We considered eight activation functions to tune: ‘*tanh*’, ‘*ReLU*’, ‘*sigmoid*’, ‘*softplus*’, ‘*softsign*’, ‘*selu*’, ‘*elu*’, ‘*exponential*’. In Appendix A, we reported the activation functions ‘*tanh*’ and ‘*ReLU*’ as the most performant functions in our datasets.

*Batch size*: Batch size is the number of training data sub-samples for the input. The smaller batch size makes the learning process faster at the expense of the variance of validation dataset accuracy. To minimize the time of the learning process as much as possible, we set the range of this value with the following values: 8, 16, 32, 64, 128.

*Number of epochs*: The number of times a whole dataset is passed through the neural network model is called an epoch. One epoch means that the training dataset is passed forward and backward through the neural network once. The number of epochs must be tuned to gain the optimal result: too few epochs typically result in underfitting, while too many epochs lead to overfitting. Hence, we verified optimal agreement of the test loss and train loss through the plot of learning curves. Following this visual inspection (see Section 3.1 and Figure 2), the number of epochs available for tuning was limited to the set: 50, 100, 150, 200.

*Lookback (k-value):* The number of time steps looked back in the prediction is a key value in multi-step ahead forecasting. The weight trend is strongly influenced by the previous values. Hence, considering previous time steps in the forecasting of the weight is necessary to reduce as much as possible the errors committed in the prediction. However, a higher value could bring unwanted results, such as decreasing the performance of the forecaster both in terms of accuracy and computational speed. Following these considerations, we considered for tuning, as a trade-off, the range: 7, 6, 5, 4, and 3.

*Dropout rate*: The dropout layer is a regularization layer. As its name suggests, it randomly drops a certain number of neurons in a layer. The dropped neurons are not used anymore. The percentage of neurons to drop is set in the dropout rate. A high value may be too severe for the application. To avoid this problem, the dropout rate was chosen from the following range: 0.2, 0.4 and 0.6 [30].

*Seasonal terms*: For each user, the seasonal term could influence the weight variation. For this reason, in the tuning, we considered whether the addition of the week cosine and week sine terms among the input variables would lead to an increase in the performance of the model or not.

Metrics and optimization algorithm: In the tuning, “*Mean Absolute Error*” (MAE) was used as the GRU loss function, and “*ADAM*” as the optimization algorithm.

### 2.7. Walk-Forward Validation and Simulation

In time series modeling, the predictions over time become less and less accurate. Walk-forward validation (WFV) is a more realistic approach consisting of continuously re-training the model with actual data as they become available for further predictions. Since the training of GRU neural networks is not too time-consuming, WFV is the most preferred solution to obtain the most accurate results.

Following the same criteria of the WFV, we defined the walk-forward simulation (WFS). The only difference between the two approaches is that in WFS, we used forecasted values as input rather than actual data. The WFS’s workflow is shown in Table 1.

A limit of WFV and WFS is the fact that the re-training phase forces the start of forecasting or simulation only from the last acquired time step. However, if there was a need to simulate effects of variations of *EB* or food composition beginning from other starting points, our approach was to avoid the re-training phase. This approach is particularly useful when input data are scarcely sampled in the training set and WFS cannot give correct responses.

### 2.8. Computer Performance

For the study, a PC with the following characteristics was used: Windows 10 Enterprise, Intel(R) Core(TM) i5-8500 CPU @ 3.00 GHz, 8 GB RAM, Intel(R) UHD Graphics 630.

### 2.9. Python Libraries

The setup used for this study was composed of the following libraries: tensorflow CPU == 2.8.0 (https://pypi.org/project/tensorflow-cpu/, accessed 5 July 2022), keras == 2.8.0 (https://keras.io/, accessed 5 July 2022), pandas == 1.0.5 (https://pandas.pydata.org/, accessed 5 July 2022), numpy == 1.22.2 (https://numpy.org/, accessed 5 July 2022), matplotlib == 3.5.2 (https://matplotlib.org/, 5 July 2022), seaborn == 0.10.1 (https://seaborn.pydata.org/, accessed 5 July 2022), pymongo == 3.11.4 (https://pymongo.readthedocs.io/en/stable/, accessed 5 July 2022) and scikit-learn == 0.24.2 (https://scikit-learn.org/stable/, accessed 5 July 2022).

## 3. Results

### 3.1. Selection of the Optimal Models through Grid Search of GRU Parameters and RMSE Overall Minimization on the Cohort of Users

As a starting point, we selected the four time series and carried out HP tuning (Section 2.6) following reduction of the RMSE of the values predicted using the test-train method with a 7-day test dataset. The optimal hyperparameters (HP) defined the individual model, called PMA, which is reported in Table 2 for each user:

We can observe from the table that the PMA differed among users with the exception of the activation function (‘*ReLU*’), the dropout rate (0.2), and the seasonal terms that gave no additional improvement to PMA. This is probably because the size of the training set spanned through a time period (i.e., winter and summer) during which well-defined habits did not arise. We also checked the test-train plots for all users (Figure 2). They showed no evident presence of overfitting, guaranteeing the goodness of the model.

### 3.2. Weight Forecasting: Model Results, WFV and WFS

In this section, we report the forecasting results of the most performant GRU for the weight forecasting and for the WFV and WFS.

The training set for the weight forecasting was selected as 90% of the overall dataset (about 330 days), yielding an RMSE averaged on the four users of 0.59±0.076.

However, predictions of 30 days, albeit with good results, could be subjected to additional uncertainty because they did not account for additional variables that could affect actual weight variations over such a long period of time (abdominal bloating due to excess food ingestion, water retention, constipation). Therefore, we carried out train-test forecasting for each user considering an interval of 7 days. The results are shown in Figure 3.

Test-train RMSE carried out with a test dataset length of one week yielded an averaged value for the four users of 0.41±0.05, showing a 30% decrease. Moreover, RMSE for each user stayed below 0.5. Despite these improved results, it is well known that in time series modeling, the predictions over time become less and less accurate (Appendix A). Therefore, WFV was the most preferred solution to obtain the most accurate results by re-training the model with actual data as they became available for further predictions. This technique could be used to perform simulations, namely WFS (Section 2.7).

The WFV and WFS for the PMA were thus performed within a week to evaluate the RMSEs with respect to the true values (Figure 4). A major improvement was obtained with this validation method, yielding an average RMSE of 0.42±0.1 for the WFV and 0.48±0.18 for the WFS. As expected, the results from the WFV were better than those from the WFS (${\mathrm{RMSE}}_{\mathrm{WFV}}<{\mathrm{RMSE}}_{\mathrm{WFS}}$). Nevertheless, the WFS showed optimal results allowing it to be used with specific applications, such as, for example, the simulation of diet plans.

### 3.3. Simulation of the Personalized Effects of Diet Plans on Weight

WFS can be used to simulate personalized diet plans and to predict metabolic responses after the introduction of new food and PA habits (determining variations in *EB* and macronutrient composition). To test the performance of the model in new simulated conditions, dietary plans were obtained by constraining the *EB* value to be constant at a particular level, and the effect of these variations on the weight of each user was simulated.

In detail, a basic simulation was carried out varying EB in the following range: −1000,−500, 0, 500, 1000 kcal (Figure 5A), with standard percentage contributions of carbohydrate, protein, and lipid intake (50/20/30%), respectively, included in acceptable macronutrient distribution ranges (AMDR) [31]. The values of macronutrient intakes were calculated by converting their percentages into grams [32]; then, the total caloric intake was evaluated by inverting Equation (1). From the simulations, we can observe that an energy deficit of 500 kcal per day yielded an average weight loss of −0.4±0.2 kg in a week, while an energy surplus of 500 kcal yielded an average weight gain of 0.77±0.63 kg in a week, and that differences existed among users. To summarize simulation results and to cancel out random effects in the daily weight variation due to water retention or constipation, we fitted the simulated trends with a parabolic fit as shown in Figure 5A and estimated the w value representing the weight value at the end of the week. In Figure 5B, individual weight variations in function of the simulated EB values are reported. These differences could be parametrized for each user by retrieving the coefficient of the relation Δw=m⋅EB+q (Table 3). Here, *q* represents the weight variation at *EB* = 0, which is, therefore, expected to be equal to zero. The *q* value can furnish an average value of eventual residual biases in data collection, yielding a systematic error in the determination of *EB*. It provided a quality factor of food insertion, which was the highest for User 2. *m* is a parameter linked to metabolic plasticity, expressed in $\frac{\mathrm{Kg}}{\mathrm{Kcal}}$, representing the rate of weight variation per unbalanced calorie. A higher value indicates a higher metabolic plasticity and/or a more active metabolism. This parameter can thus be used to develop a metabolic taxonomy of the users. In our use case, users 0 and 2 showed higher metabolic plasticity than users 1 and 3.

### 3.4. Personalized Diet Plan: Use Case

In this use case, rather than performing a toy diet plan, we tested an actual personalized diet plan on User 2 to achieve weight loss in a healthy way supervised by a professional nutritionist considering blood analyses, food and activity habits.

In Figure 6, the actual weight variations (black), the prediction made by WFS using as exogenous data the actual data (red), and the WFS using as exogenous values the data retrieved from the personalized diet plan (green) are shown. The RMSE of the prediction was 0.26 (showing that the technique had good performance), and the weight loss approximately of Δw=1.5 kg following the tailored diet plan provided to the user, which was in accordance with the predetermined goal defined by the nutritionist (rapid weight loss). This tool can thus allow us to compare the expected and actual effects of the diet on the weight variations and to test several nutritional plans in terms of energy balance and macronutrient composition.

## 4. Discussion

Obesity and its metabolic complications are the most serious public health challenges of the 21st century. The prevalence of obesity has tripled in many countries of the EU [33]. In the current pandemic, the issue of obesity has become more prominent [34], highlighting the need for its prevention. Evidence that relates to obesity is biased towards its causes rather than strategies for prevention, which have not yet been widely replicated or delivered at a scale offering clear options for public health strategies. Finding and implementing solutions require new models able to implement healthy lifestyles and prevent illness by relying on devices that can be used in daily life, reducing the burden on hospitals. Here, we relied on an application able to retrieve, pre-process and analyze spontaneous and voluntary PA, diet, and anthropometric quantities from a set of wearables and home-portable devices provided to the end-user. These data drove the development of a personalized model of the end-user metabolism, the PMA, able to estimate his/her personalized reactions to diet, PA, and environmental and psychological factors. The PMA was integrated into the IoT-reliant infrastructure, allowing it to perform simulations and predictions to gradually improve adoption of healthy habits.

In this manuscript, we have shown how GRU-based deep neural networks are a good solution to predict in an accurate way the weight for the day after (the WFV showed an average RMSE lower than 0.5), and to simulate personalized diet plans to help reach ideal weights in an healthy way, avoiding excessive variations in habitual diet or PA and keeping weight and nutrient balance in the normal range following guidelines. We tested the PMA by using WFS to predict the weekly weight variations of four users subjected to varying energy balance constraints, and we also converted a true nutritional plan developed by a professional nutritionist in a WFS to test the effect on a user, with the aim of evaluating if the metabolic response of the subject could achieve weight loss.

The principal strength of the PMA with respect to established knowledge-driven models resides in the fact that the developed data-driven model can take into account all of the processes involved in metabolism having an influence on weight variations, from genetic predispositions to current microbiome composition. Nutrigenomics (also known as nutritional genomics) is broadly defined as the relationship between nutrients, diet, and gene expression [35] having a deep influence on individual metabolism [36]. ‘Microbiome’, also called ‘gut microbiota’ [37], is a complex and dynamic population of microorganisms that exert a marked influence on the host metabolism during homeostasis and disease. Multiple factors contribute to the establishment of the human gut microbiota during infancy, and diet is considered as one of the main drivers in shaping the gut microbiota across one’s lifetime. The data-driven nature of the PMA allows it to integrate the complexity of these metabolic processes without the requirement of deterministic or statistical models, which make generalizable claims in trying to describe human metabolism for all human subjects, or for certain subsets of the population. If this is the objective, the distribution needs to be accurately sampled from the population on which the claim is made, and the number of subjects has to be adjusted to improve the significance of the prediction. Here, the claim is different, because we did not realize a single general model, but four distinct models of metabolism, personalized for each individual. We modeled personal metabolism as a black box in which the input was energy balance and macronutrient composition, and the output was weight. In this framework, the statistical unit, rather than the subject, is the daily response of individual weight to the different input stimuli. This allowed us to make forecasts based on a high number of available data (~300 per person). We were able to gain a feel for these peculiar PMA features by comparing its performance with available weight predictors [11]. Nowadays, available weight predictors use general information such as age, sex, height and current weight to forecast weight variations by setting a predefined value of energy balance. As shown in Table 4, these types of data, based on a statistical model describing average features of the analyzed sample population, intrinsically do not allow an actual personalized prediction. The PMA instead allows descriptions of personalized metabolic responses for users, as quantified by the standard deviation of the predictions (0.2 kg), which is almost 10 times that of weight predictors (0.034 kg).

In Figure 7A,B, we can observe how the statistical model (blue points) predicts a weekly weight loss for an EB=−500 kcal, which shows slight variations with starting BMI, age or sex. While Users 0, 1 and 3 were well aligned with the general population, we observed that User 2 deviated from the general trend. This was indeed the subject with the highest metabolic plasticity in the systematic simulation performed in Section 3.3.

These anomalous values of metabolic plasticity can be due to several factors, ranging from microbiome diversity to a different nutri-genotype. Additionally, hormonal equilibria and systemic diseases can have a huge influence [38]. This difference with the general trend highlights how a personalized approach, in this particular case, is fundamental in assessing tailored weight loss in response to nutritional treatments. A correction of the metabolic plasticity with microbiome composition and diversity or with nutrigenomic characteristics would be an important advancement in understanding the factors leading to the reshaping of individual metabolism. The clinical relevance of the results presented in the manuscript resides in the possibility to understand if metabolic adaptations due to microbiome variation or general metabolism reprogramming due to treatments or nutritional interventions are occurring, and how to change them through simulations in order to fulfill desired results. Applications can be envisioned for obesity and nutritional disorder treatments, and to generate diet plans in synergy with treatments in cancer and other diseases.

Other than personalization, an additional strength of the PMA resides in the informative content of the inputs: information such as food composition allows better prediction of the metabolic response. Indeed, to reach ideal goals such as weight loss, a correct subdivision of the basic nutrients is fundamental in the generation of a diet plan. It is in principle possible also to include other important variables, going from micronutrient composition of the diet and the use of integrators to sleep quality.

The PMA is also scalable not only in terms of its inputs, but also in terms of outputs, allowing it to contextually predict changes in variables of interest other than weight (e.g., fat and lean mass, resting heart rate).

Therefore, the PMA could become a powerful support tool for nutritionists, dieticians, physicians, etc. Hence, it has the potential to lay the foundations for truly ‘personalized nutrition’ approaches, using these predictions to identify metabolic impairments and plan actions in advance, and to simulate the metabolic response to several diet plans to achieve the desired results without compromising the body’s wellness. The generated diet and activity plans could be delivered to users by front-end components with a virtual assistant helping patients to monitor their behavior and improve their adherence to optimal actions. However, the PMA has some critical issues. First, the prediction of weight with unknown conditions, such as for extreme diet plans (e.g., the ketogenic type), could lead to inaccurate predictions because the PMA may lack training on that data. The PMA could overcome these problems by relying on continuous training day after day. Moreover, noise caused from wrong data insertion could alter the quality of predictions. Another point is that, at the current stage of development, the PMA requires data collection for at least 2 months to achieve good performance. To improve data collection, automatic food detection methods through mobile phones [39], which are continuously evolving, could overcome this limit by reducing manual compilation and decreasing the burden on users.

## 5. Conclusions

This study shows that the integration of several IoT devices and a diet registry into a single web app able to merge all acquisitions into a single visualization dashboard, with a deep learning analysis of user metabolism through the realization of PMA, provides important information to realize optimal weight forecasting and the personalized generation of diet and activity plans. Relying on this information, appropriate clues can be obtained to empower citizens with scientific knowledge and validated instruments, augmenting their self-awareness with the aim to achieve long-lasting results in the pursuit of a healthy lifestyle. An important advancement could be the integration, as input in the PMA, of novel developed biomarkers of lipid metabolism (such as membrane lipids and membrane fluidity of red blood cells) to study the effects and influence of dietary molecules on their outcomes [40,41,42,43,44]. Moreover, innovative and promising anthropometric markers tracked with wearable devices, such as VO_2_max and heart rate variability (HRV), can improve the performance of weight forecasting [45,46,47]. These integrations could explain and cluster the different responses given by the PMA, furnishing insights into the factors able to shape individual metabolism.

## Figures and Tables

**Figure 1 nutrients-14-03520-f001:**
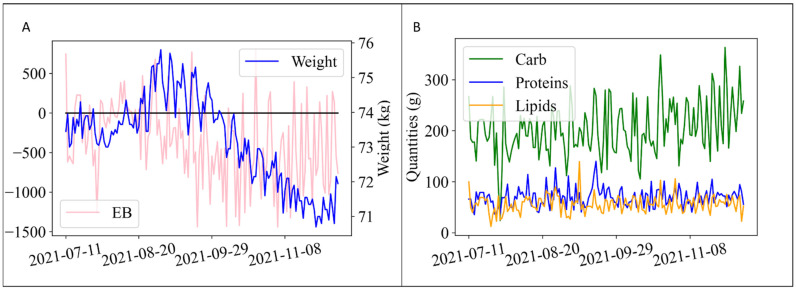
Time series describing user metabolism. (**A**) Representative time series for weight and EB. (**B**) Representative time series for food composition.

**Figure 2 nutrients-14-03520-f002:**
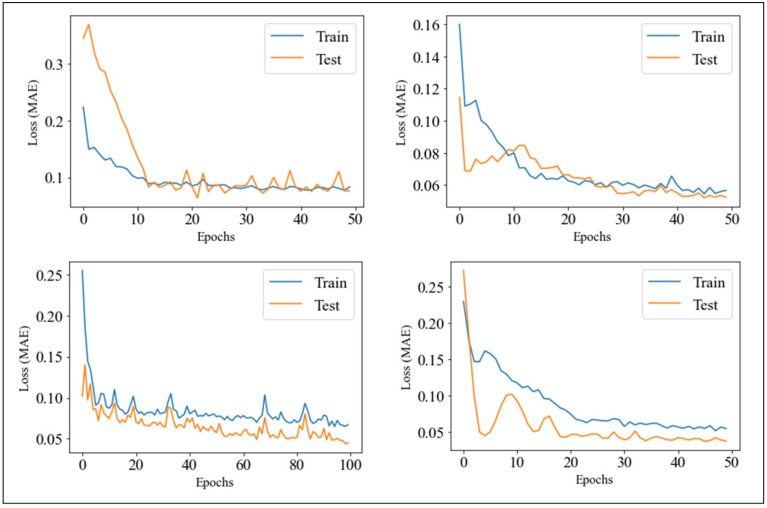
Train and test loss function (*Mean Absolute Error*) versus the number of epochs.

**Figure 3 nutrients-14-03520-f003:**
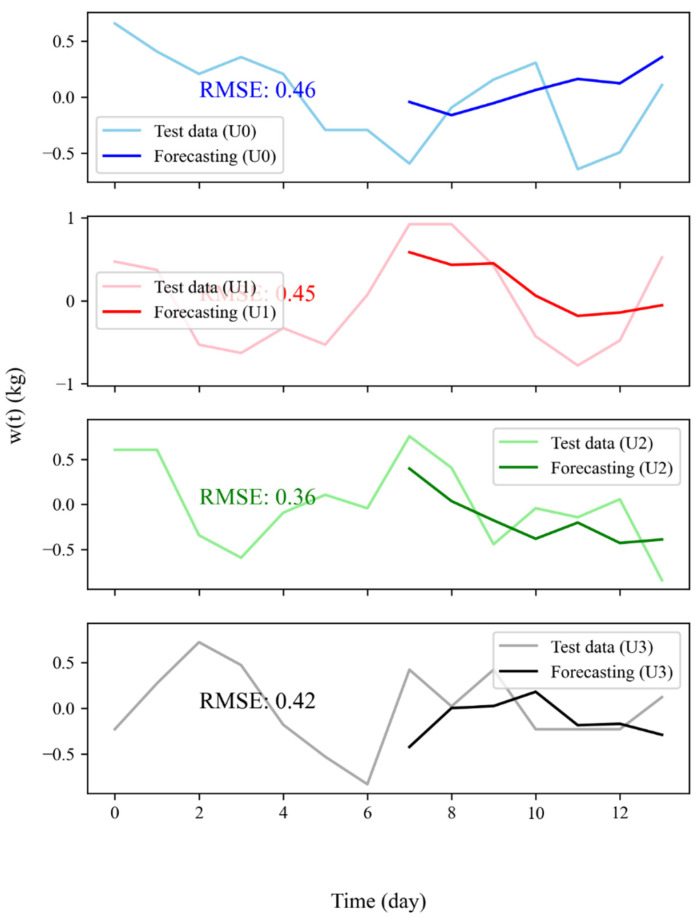
Test-train forecasting for all users (U0, U1, U2 and U3) with the relative root mean squared value.

**Figure 4 nutrients-14-03520-f004:**
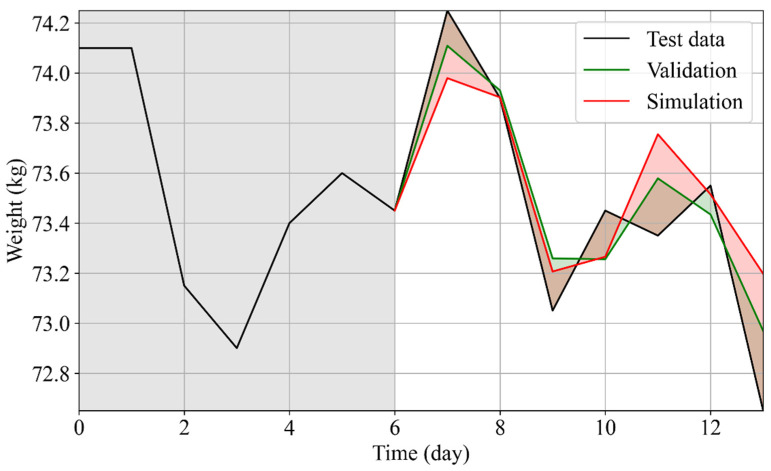
Comparison between actual data and WFV and WFS results for User 2.

**Figure 5 nutrients-14-03520-f005:**
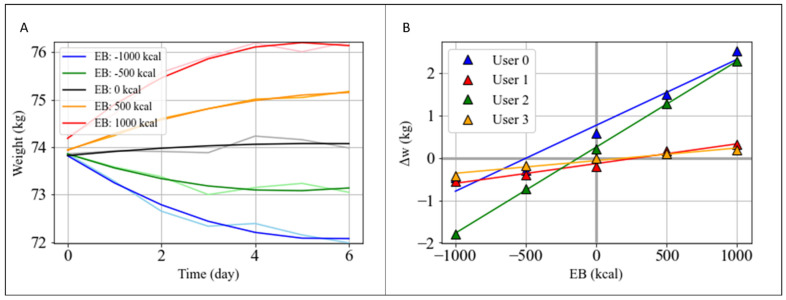
Effects of diet plans on user metabolism. (**A**) WFS performed at different EB values on the data of User 2, keeping constant the percentage of macronutrient intake (50%, 20%, 30%, respectively). Weight data were fitted with a second order polynomial. (**B**) Weight variation Δw calculated from the first and last values of the fit of the second grade versus the EB value and for each user.

**Figure 6 nutrients-14-03520-f006:**
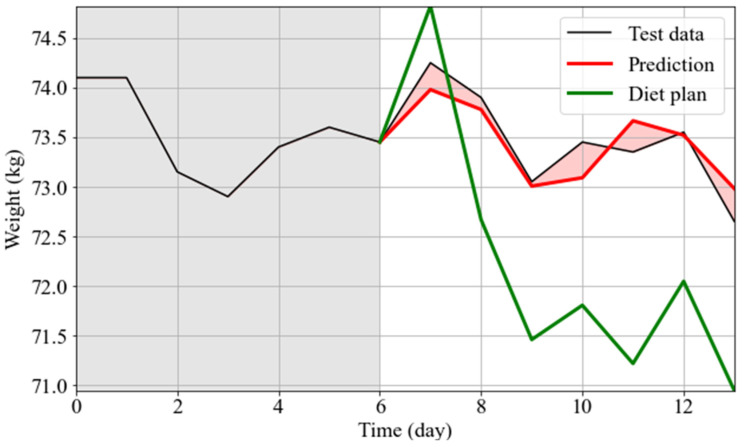
Personalized nutritional intervention plan for User 2. In the first 7 days, the actual weight trend is shown (black line, gray shaded area). Along this trend, WFS for the personalized plan is reported (green line). As a control, WFS when covariates retained the actual values is reported (red line).

**Figure 7 nutrients-14-03520-f007:**
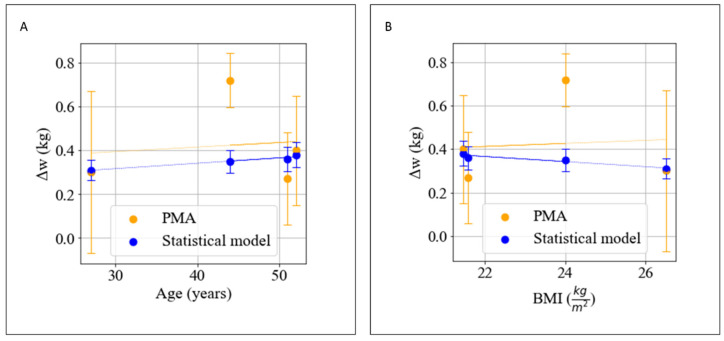
(**A**) Δw calculated with PMA and with statistical models with respect to age of users. (**B**) Δw calculated with PMA and with statistical models with respect to BMI of users. For the statistical model, an error of 15% was considered, while for PMA, it was considered as an error of the RMSE of the WFS.

**Table 1 nutrients-14-03520-t001:** Concept of WFS.

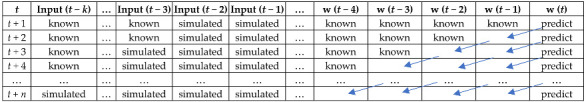

Columns represent input values at time t. Input (*t − k*); …; Input (*t* − 1) represent covariates, while w (*t* − *k*); …; w (*t*) represent the target variable (weight). Rows represent predictions at time *t* + 1, *t* + 2, …, *t + n*. ‘known’ means that the value is taken from the dataset of actual values, ‘simulated’ indicates that the value is an input of a simulated diet plan, ‘predict’ indicates that the value is predicted from the neural network.

**Table 2 nutrients-14-03520-t002:** Results of the hyperparameter tuning for each user.

User	Number of Neurons	Activation Function	Dropout Rate	Epochs	Batch Size	Lookback	Seasonal Terms	RMSE
0	100	*ReLU*	0.2	50	32	7	No	0.47
1	200	*ReLU*	0.2	200	128	4	No	0.49
2	150	*ReLU*	0.2	50	64	5	No	0.31
3	100	*ReLU*	0.2	50	128	5	No	0.4

**Table 3 nutrients-14-03520-t003:** Metabolic plasticity m and quality factor q for each user.

User	$\mathrm{Metabolic}\;\mathrm{Plasticity}\;(m)\;[\frac{\mathrm{kg}}{\mathrm{kcal}}]$	Quality Factor (q) [kg]
0	1.56·10^−3^	0.77
1	0.47·10^−3^	−0.13
2	2.03·10^−3^	0.26
3	0.30·10^−3^	−0.06

**Table 4 nutrients-14-03520-t004:** Comparison of weight predictions between statistical and data-driven models (PMA).

User	Age	Sex	Height (cm)	*w_i_* [kg]	∆*w* (PMA) [kg]	∆*w* (Statistical Model) [kg]
0	27	M	183	88.7	0.3 ± 0.37	0.31 ± 0.031
1	52	M	186	74.25	0.4 ± 0.25	0.38 ± 0.038
2	44	M	175	73.45	0.72 ± 0.12	0.35 ± 0.035
3	51	F	160	55.25	0.27 ± 0.21	0.4 ± 0.04

## Data Availability

Data and codes are available upon reasonable request at https://github.com/Metabolicintelligence/PMA, accessed on 25 August 2022.

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
