# Peer review of "Personalized Metabolic Avatar: A Data Driven Model of Metabolism for Weight Variation Forecasting and Diet Plan Evaluation"

_nutrients, 2022, doi:10.3390/nu14173520_

Round 1

Reviewer 1 Report

The manuscript in this form contains a lot of jargon suitable for a journal on medical informatics. For readers who are clinicians, this data is difficult to comprehend. 

The need for the study is not justified in the introduction. Much of the data from these devices is already known. This appears using complicated algorithms on a handful of four subjects only. 

The clinical relevance of the graphs is not clear. 

Author Response

Dear Reviewer,

Thank you for your kind message concerning our manuscript entitled “Personalized Metabolic Avatar: a data driven model of metabolism for weight variation forecasting and diet plans evaluation”.  We thank you  for the time and effort put into assessing the manuscript. We carefully considered your comments and improved the manuscript. Below is our point-to-point response to the comments. 

Sincerely,

 Prof. Giuseppe Maulucci, PhD

 ----------------------------------------------------------------------------

Reviewer 1

The manuscript in this form contains a lot of jargon suitable for a journal on medical informatics. For readers who are clinicians, this data is difficult to comprehend.

We thank the reviewer for raising this issue, we simplified or explained better some terms in order to improve the readability of the manuscript, by leaving technical details essentially in the materials and methods section. In the supplementary materials an extended description of the model is furnished. 

 The need for the study is not justified in the introduction.

We agree with the reviewer that the improvements are not sufficiently explained in this section. As we stated in the introduction, the problem of the prediction of weight variations traditionally relies on estimations based on thermodynamic models depending on age, height, gender, and current weight [11]. However, diet predictors developed through these models have limited application because they assume weight stability and do not account for factors like microbiome, variations in type and expression of genes linked to nutrition,  and quality and quantity of physical activity.  Differently from the well-established knowledge-driven models, data-driven models can account for all the metabolic processes, from genetic predispositions to current microbiome composition, affecting weight changes. Relying on this information, data driven models could offer crucial data for achieving the best weight forecasts and the creation of individualized diet and exercise plans with the aim to achieve long lasting results in pursuing a healthy lifestyle.  We added a paragraph in the introduction to address this issue (lines 72-78).

 Much of the data from these devices is already known. 

The interface to acquire in real time the data was already developed and published. Relying on these data, the purpose of this article is instead to develop a personalized model of metabolism. We hope the paragraph added in the introduction helps to clarify this point.

This appears using complicated algorithms on a handful of four subjects only. 

We thank the reviewer for raising this point. This is very appropriate in this context since usually the models make generalizable claims in trying to describe human metabolism for all the human subjects, or for certain subsets of the population. If this is the objective, the distribution needs to be accurately sampled from the population on which the claim is made and the number of subjects has to be adjusted to improve the significance of the prediction. Here the claim is different, because we did not realize a single general model valid for all individuals, which would require an extension of the sample number, but four models of metabolism, each one personalized for the subject under investigation. We modeled personal metabolism as a black box in which the input is energy balance and macronutrient composition, and the output is weight. In this framework, the statistical unit, rather than the subject, is the daily response of individual weight to the different input stimuli. This allows us to make forecasts based on the high number of available data (i.e. 300 per person). In the future, we plan to assemble a single model of human metabolism, rather than several models. Obviously in this case, in order for the model to work, we will need to include more inputs, putting out of the black box hidden variables,  and to raise the number of individuals. We added a paragraph to discuss this point (lines 454-464)

The clinical relevance of the graphs is not clear. 

We thank the reviewer for raising this point.  The clinical relevance of the results presented in the manuscript resides in the possibility to understand if metabolic adaptations due to microbiome variation or general metabolism reprogramming due to treatments or nutritional interventions are occurring, and how to change them through simulations in order to fulfill desired results. The applications can be envised in obesity and nutritional disorder treatments, and to generate diet plans in synergy with treatments in cancer and other diseases. we added a paragraph to explain better which is the relevance of these graphs (lines 492-497).

Reviewer 2 Report

The work is well designed. However, the significant deficit of the study is the tiny number of subjects included in the study. Only 4 individuals were included in the study. How was this number determined? I think at least 12 individuals should be included in the study. I recommend that reviewers review the article again after the authors expand the study. On the other hand, if the authors can show reliable evidence that 4 individuals are sufficient, I have no objection to the study being accepted.

Author Response

Dear Reviewer,

Thank you for your kind message concerning our manuscript entitled “Personalized Metabolic Avatar: a data driven model of metabolism for weight variation forecasting and diet plans evaluation.”.  We thank you  for the time and effort put into assessing the manuscript. We carefully considered your comments and improved the manuscript. Below is our point-to-point response to the comments. 

Sincerely,

 Prof. Giuseppe Maulucci, PhD

-------------------------------------------

Reviewer 2

The work is well designed. However, the significant deficit of the study is the tiny number of subjects included in the study. Only 4 individuals were included in the study. How was this number determined? I think at least 12 individuals should be included in the study. I recommend that reviewers review the article again after the authors expand the study. On the other hand, if the authors can show reliable evidence that 4 individuals are sufficient, I have no objection to the study being accepted.

We thank the reviewer for raising this point. This is very appropriate in this context since normally the models make generalizable claims in trying to describe human metabolism for all the human subjects, or for certain subsets of the population. If this is the objective, the distribution needs to be accurately sampled from the population on which the claim is made and the number of subjects has to be adjusted to improve the significance of the prediction. Here the claim is different, because we did not realize a single general model valid for all individuals, which would require an extension of the sample number, but four models of metabolism, each one personalized for the subject under investigation. We modeled personal metabolism as a black box in which the input is energy balance and macronutrient composition, and the output is weight. In this framework, the statistical unit, rather than the subject, is the daily response of individual weight to the different input stimuli. This allows us to make forecasts based on the high number of available data (i.e. 300 per person). In the future, we plan to assemble a single model of human metabolism, rather than several models. Obviously in this case, in order for the model to work, we will need to include more inputs, putting out of the black box hidden variables,  and to raise the number of individuals. We added a paragraph to discuss this point (lines 454-464)

Round 2

Reviewer 1 Report

Thankyou for the corrections 

Reviewer 2 Report

The authors made the necessary explanations. I have no further comments.